# Urban Blue Spaces as Therapeutic Landscapes: “A Slice of Nature in the City”

**DOI:** 10.3390/ijerph192215018

**Published:** 2022-11-15

**Authors:** Niamh Smith, Ronan Foley, Michail Georgiou, Zoë Tieges, Sebastien Chastin

**Affiliations:** 1School of Health and Life Sciences, Glasgow Caledonian University, Glasgow G4 0BA, UK; 2Department of Geography, Maynooth University, W23 HW31 Kildare, Ireland; 3School of Computing, Engineering and Built Environment, Glasgow Caledonian University, Glasgow G4 0BA, UK; 4Department of Movement and Sports, Ghent University, Watersportlaan 2, 9000 Ghent, Belgium

**Keywords:** blue space, urban environment, therapeutic landscape, qualitative data

## Abstract

Urban blue spaces are defined as all natural and manmade surface water in urban environments. This paper draws on how the concepts of experienced, symbolic, social, and activity space combine to position urban blue spaces as therapeutic landscapes. We conducted 203 intercept interviews between 12 October 2019 and 10 November 2019. Although safety concerns had health-limiting impacts, interacting with the Glasgow Canal and surrounding landscape was predominantly perceived as health-enhancing. Our findings build on current evidence, which has suggested that urban blue spaces, particularly canals, may foster therapeutic properties, contributing to healthier city environments. Further research is required to understand better the interconnectedness of urban blue spaces and health and how such spaces can be best developed and managed to improve the health outcomes of local populations.

## 1. Introduction

The intersections of place and health are researched across multiple disciplines, including geography, psychology, epidemiology, public health and anthropology. Within health geography, the term ‘therapeutic landscape’ has been used to understand how different environments affect the experiences of health and contribute to a healing sense of place [1]. Initially, the concept was applied explicitly to exceptional places of healing such as pilgrimage sites, hot springs, baths and spas, but over time it has been adopted more widely to encompass everyday places that promote and maintain health and wellbeing [2]. Specific natural, built, social and symbolic environments can be considered therapeutic landscapes [3]. Therapeutic landscapes can contribute more widely to interdisciplinary research into healthy spaces, places, and related practices [4]. The language around therapeutic landscapes has evolved to place attention on green and blue spaces and both the positive and negative potentials of different spaces [4]. Volker and Kistemann (2011) developed a framework to analyse blue spaces as therapeutic, drawing on the core material, social, spiritual and symbolic dimensions [5]. Because blue spaces are recreational activity sites, and due to the strong correlations between activity and health, they added an ‘activity’ dimension within the concept of therapeutic landscapes [6]. This framework is further detailed in the Methods below as it is used to guide our data analysis.

Cities have traditionally been built around coastlines and along natural and purpose-built waterways to aid trade and commerce. Historically, urban blue spaces were sites of industry, dominated by ports, shipbuilding and commercial transportation. Following the decline of heavy industry in many European and other developed countries worldwide and changes in transportation and manufacturing, many urban waterways have become redundant and neglected [7]. More deprived and marginalised populations often end up living along these rundown, disused urban inland waterways, and so understanding how urban blue spaces could benefit urban residents can address these environmental injustices [7].

More recently, urban blue spaces have been reclaimed and redeveloped to exploit the synergies between innovative urban design, social and economic advancement and historical and cultural identity to return benefits to urban residents and increase access to natural spaces in cities. These redevelopments and the reclamation of underutilised urban blue spaces present potential innovative solutions to urban problems. Many of these have been explored within academia, including acknowledging urban blue spaces in response to the threats of climate change [8], socio-economic inequality [9], economic stagnation [10] and society’s disconnection from nature [11]. Waterfront properties, including those with views of the coast, river estuaries, harbours and inland waterways, are in increasingly high demand; they are valued at 51% more than homes located a mile from water with no “blue view” [12]. However, it is critical to note that the development and rise in demand for waterside properties could contribute to “blue gentrification”, where exclusive housing developments displace those living close to the blue spaces and care must be taken to mitigate such impacts [9].

Interest in the intricate relationships between urban blue space and public health is growing within academia. Historically, water environments, including springs and holy wells, have been the sites of healing and wellness [13], and people continue to value the healing properties experienced by water through visits to spa towns, baths, sites of pilgrimage as well as more modern spas and hydrotherapy [14]. However, water has also been risky and health-limiting [7,15] as contaminated water can spread disease [16] and there is also a risk and fear of drowning [17]. More recently, there has been a shift in focus towards the health-promoting capacity of blue spaces, as seen in the growing body of literature [6,18,19,20,21].

Research on the health-promoting capacities of local urban canals is emerging. Retrospective longitudinal studies of the canals, looking at a macro-geographical level, found that the regeneration of Glasgow’s canals was associated with a 3% decrease in mortality rate [22], 10–15% decrease in risk of chronic diseases [23] and 6% decrease in the effect of deprivation on the risk of mental health issues [24]. Qualitative research explored locals’ experiences of the Niasarm Canal in Isfahan, Iran, concluding that it is an active space that promoted feelings of relaxation and concentration and was also seen to foster identity among users and shaped the canal as a therapeutic landscape [25]. Other researchers have moved away from the term ‘blue space’ to think more about ‘wateriness’ to capture the complexity and variety of canals [7]. Additionally, research on canals in Leeds, England discusses water as a potential agent of care [26]. They studied the concept of ‘hydrocitizenship’, examining how water can benefit people but how simultaneously, people can develop greater respect for water and develop pro-environmental behaviours, often linked to regular exposure [26].

The above studies on canals collectively identify specific ‘linear geography’ that speaks to a sense of waterways acting as mobile and go-along health assets. Linear parks may allow more equitable access to green space than those with more compact shapes [27], and it has also been found that people predominantly use linear parks for exercise [28]. The same may be true for ‘linear blue’ spaces and they may be more equitable therapeutic landscapes. Our research builds on these previous studies of the beneficial health impacts of urban blue spaces to explore them as a ‘slice of nature in the city’. The research question we aimed to address was, “How does the Glasgow Canal work as a therapeutic urban blue space?” We argue that urban blue spaces, such as canals, can contribute to health-enabling urban environments as examples of therapeutic landscapes particularly due to their linearity and their recognition as everyday spaces.

## 2. Methods

This study adopted a qualitative design using intercept interviews, where respondents were approached to participate in a short interview to capture responses in-situ on the canal [29]. Following Consolidated Criteria for Reporting Qualitative Research (COREQ) guidelines [30], this study reports results from 203 intercept interviews completed with users of the canal between 12th October 2019 and 10th November 2019.

### 2.1. Study Site

We collected data at three access points to the Glasgow Canal, part of the Forth and Clyde Canal. The Forth and Clyde Canal closed to navigation in 1963. During this time, the canal was fenced off and was an abandoned ‘excluded blue space’ [31]. Since the early 2000s, the canal has been under regeneration as part of the Millennium Link Project. Today, it serves as an area of recreation in the city. The canal is surrounded by some of the most deprived areas in Glasgow, as defined by the Scottish Index of Multiple Deprivation (SIMD) (Figure 1). High levels of deprivation are, in part, a result of the legacy of de-industrialisation in North Glasgow. We chose three sites for data collection: Stockingfield Junction, Maryhill Road and Applecross, to understand the canal’s health-enhancing and health-limiting properties across multiple geographies (Figure 2). These sites were chosen as they were geographically spaced and were access points to the canal towpath, so they were likely to have more footfall. The sites were within approximately 20 min walking distance from one another, and so capture entry and exit points on typical local walks or commutes into the city centre.

In Glasgow, at the time of data collection, temperatures range from an average minimum of 3.55 °C to an average maximum of 12.75 °C and it rains on 17 days/month on average [32].

### 2.2. Participants

Adults were recruited through purposeful convenience sampling as they walked by the three study sites and a similar number of participants were interviewed at each site. We wanted to understand the perceptions of canal users who were actively in the space [33]. Intercept interviews are commonplace in consumer research and are starting to be used more frequently in academia to explore perceptions of outdoor environments [34,35,36]. The fast-paced nature of modern life means that people may generally be too busy to engage in lengthy research [29]. Therefore, intercept interviews allowed us to increase the feasibility of data collection as we were able to recruit in situ and carry out brief interviews without disrupting too much of our participants’ day. A researcher approached potential participants, explained the study and invited them to take part immediately. Potential participants were informed that participation was voluntary and that the interview would take approximately five minutes. Before taking part in intercept interviews, participants were read information about the study and provided informed consent verbally. Participants were not compensated for their time. Recruitment occurred until data reached saturation [37]. Ethical approval was obtained from the School of Health and Life Sciences at Glasgow Caledonian University (ethical approval code: HLS/PSWAHP/17/144),

### 2.3. Data Collection

Seven researchers conducted intercept interviews (four female and three male). Interviewers were trained in qualitative data collection and had no existing relationships with any participants. There were also no conflicts of interest or bias. We conducted intercept interviews modelled on previous studies investigating the salutogenic impact of urban blue space [25,38,39] and used the Epicollect 5 mobile data collection app to develop a questionnaire and document responses [40]. Firstly, researchers noted the date, time and study site in the app. We then asked participants to provide brief demographic data including gender, ethnicity, age, and disability status. There were then closed multiple-choice questions asking how often participants visited the canal and how long they spent there. The next question was audio recorded and asked what would make them visit more. They were asked about the purpose of the visit today and what their thoughts were on the accessibility of the canal. Participants were asked to rate their general health from excellent to poor [41]. Finally, participants were asked how they thought being on the canal affected their physical and mental health before being asked for their postcode and length of time at that address. Additional prompts were used when required, such as “Do you want to say a little more about that?”. All interviews were conducted in English and averaged five minutes in duration (range: 1 to 13 min).

### 2.4. Conceptual Framework:

We use Völker and Kistemann’s (2011) conceptual framework to examine the relationships between urban blue space and health, focusing on the Glasgow Canal as a case study. The framework draws on the concept of therapeutic landscapes, acknowledging the core material, social, spiritual and symbolic dimensions initially postulated by Gesler and Kearns (2005) [5]. Because blue spaces are recreational activity sites, and due to the strong correlations between activity and health, Völker and Kistemann (2011) added an ‘activity’ dimension within the concept of therapeutic landscapes. Activities can be passive or active. People can, therefore, interact with space in four ways, through experienced spaces, activity spaces, social spaces and symbolic spaces [6]. People’s health can be both positively and negatively affected within these dimensions through health-enhancing and health-limiting factors.

The above dimensions refer to how people use and experience urban blue spaces. Within each of the four dimensions of appropriation, we can look at how the urban blue spaces themselves can be analysed. Völker and Kistemann (2011) categorise these dimensions as naturalistic, built, humanistic, structuralist and post-structuralist [6].

### 2.5. Data Analysis

Each participant was ascribed a number identifier to allow for anonymous reporting. These identifiers are in superscript following quotations and ideas in the findings section below. Audio files were transcribed using intellectual verbatim transcription by the seven researchers. Transcripts were read thoroughly, and preliminary semantic thematic coding took place inductively by one author [42] in NVivo 12 [43]. Initial codes were refined, and related ideas were combined to create themes [42]. At this point, a second researcher coded a subset of transcripts and disagreements were discussed to reach consensus. A second round of coding involved using Völker and Kistemann’s (2011) conceptual framework to conduct deductive Framework analysis to interpret the health-related experiences of people on the canal [6]. Framework analysis allows the researcher to classify and organise data according to deductively derived key themes and subthemes to conduct cross-sectional analysis using a combination of data description and abstraction [44]. The initial codes were categorised within the four dimensions of blue space appropriation: experienced space, activity space, social space and symbolic space, while considering the naturalistic, built, humanistic, structuralist and post-structuralist dimensions [6]. Quasi-statistics were included to make statements such as “some”, “many” and “a few” more precise [45].

## 3. Results

Between 12 October 2019 and 10 November 2019, 203 participants partook in intercept interviews; 71 at Stockingfield Junction, 80 at Maryhill Road and 52 at Applecross. The demographic descriptors of the participants are detailed in Figure 3. There was a fairly even representation of participants interviewed across the three sites: 71 at Stockingfield Junction, 80 at Maryhill Road and 52 at Applecross. Of the 203 participants who agreed to complete the intercept interview, 109 identified as male and 93 as female and the majority (91%) of participants identified as ‘White/Scottish/British’. Seventy-one participants were aged between 18 and 34, 99 were between 35 and 64 and 33 participants were 65 or over, and 6% of the sample considered themselves to have a disability.

Although the three study sites have subtle differences in terms of the level of green space, distance to a road, and interrupted views (Figure 2), there were no notable differences in responses across the sites. This suggests that the therapeutic benefits of blue space existed, irrespective of the nuances of the environment.

Through framework analysis, we identified the powerful ways urban blue spaces can promote health and wellbeing as experienced, activity, social, and symbolic spaces. A summary of themes and how they were attributed to the framework is detailed in Figure 4.

### 3.1. Experienced Space

#### 3.1.1. Fresh Air

How people experience and perceive urban blue spaces affects their health-promoting capacity. Twenty-two participants (11%) commented on experiencing the fresh air of the canal. Others expanded on this, explaining how they are not breathing in the fumes ^65, 93^; the canal is a “natural artery” ^77^ and “a bit like a lung” ^85^.

#### 3.1.2. Water As (Un)appealing

The lure of water environments has been studied at length within environmental psychology [6]. Seven canal users admired the “running water…a nice, scenic backdrop” ^52^. Others specifically identified the water as something which positively affects their mental health:“I love the water; it makes me feel a bit more alive and energetic” ^118^.“for my mental health, just being able to walk alongside the water is incredible” ^132^.“the water’s really relaxing” ^109^.“the sound of the water really helps to calm me down” ^147^.

While these individuals felt positive about the water, another said, “I don’t know if I’m overly impressed with the darkness of the water. I’ve never liked looking at the water” ^42^. Here, the health-limiting aspect of waterscapes is apparent as spaces can simultaneously be considered relaxing and risky by different people [47].

#### 3.1.3. A ‘Slice of Nature’

“It’s hard to believe you’re in Glasgow sometimes! You get away from your city, but you’re actually still in your city, in the middle of it” ^10^.

A further 17 participants (9% of sample) echoed this sentiment. They experienced the duality of urban life and the natural environment, with the canal being “so easily accessible but actually really good to get away from the busyness” ^84^ and experience “a slice of nature in the city” ^130^. Participants expressed their love for living within the vibrant, bustling city of Glasgow but highlighted “it’s important to escape” ^118^; “I think being in a city, the canal is something that you need” ^3^. “Something like this being in the heart of the city is amazing” ^139^.

#### 3.1.4. Blurred Green and Blue

The canal was poetically described as “a green corridor through the city” ^42^, conjuring a vivid image of the significance of the canal and its surrounding greenery. Thinking about the canal as a ‘green space’ is significant. Four participants (2%) referenced the canal as a green space or a park: “I mean I think Glasgow is good for parks… I think they’re really, really important” ^85^ with one participant explicitly making the connection between outdoor natural space and health:“I think it’s important to have green space in the city for peoples’ mental health and wellbeing” ^98^.“the canal has a greener sort of feeling than walking through the city” ^199^.

#### 3.1.5. Wildlife

Human engagement with the natural environment and non-human actors is a crucial feature of urban life. Five participants (2%) commented on enjoying the wildlife along the canal. Seven participants (3%) mentioned enjoying the birds, with one explicitly commenting on the bird “whistling” ^101^. Swans are a ubiquitous feature of the Glasgow Canal landscape, with one participant delighting in seeing swans “flying along the canal and landing”, exclaiming, “what more could you want?!” ^185^. Another participant wanted to see “more ducks in the water” ^66^ as they enjoy feeding them, while one more suggested having signs about the native wildlife and plants would benefit the area and educate individuals on the animals and their habitats ^140^.

#### 3.1.6. Seasonality and Weather

The natural environment visually changes over the passing of a year. Participants enjoyed experiencing the different seasons through the canal environment:“you see the seasons, so you see the swans and the nests, and then the swans’ eggs hatching” ^14^.“it’s really nice, especially when the swans have just had their little cygnets, and I got to watch them grow up” ^181^.

Ten participants (5%) commented on how the weather changed their canal experience and how they felt about using it. However, the weather added to the experience of some users:“being out in the fresh air as opposed to being stuck in a gym somewhere on a treadmill or on a bike. The canal is far better cause you’re out here in all winds and weathers” ^155^.“you can be out here in the wind and the rain, and when you go home, you know you’ve been out in the wind and the rain. It definitely makes a difference” ^65^.“it’s good to be able to go out in all weathers and try to stay fit and active” ^132^.

These comments suggest that weather can indirectly affect health and wellbeing, with some preferring to exercise in better weather. Although some participants saw the weather as health-limiting, others thought it added to their experience, making it health-enhancing.

#### 3.1.7. Quietness

Participants enjoyed experiencing the quietness of the canal, which allowed one person to “listen to my music quite happily” ^148^ and two others described it as “peaceful” ^70, 135^. Spaces are often experienced and negotiated differently by different sections of the population. One participant highlighted that the quietness was “better (than the road) for a deaf person like myself” ^185^.

#### 3.1.8. Traffic-Free

The urban and natural space dichotomy was further explained by participants using the canal path instead of the busy city roads for leisure and commuting. This predilection was described as “a no brainer” ^8^; the canal path is the obvious choice. It provides an area free from traffic, where “you’ve got the calm you can switch off and just take in the nature” ^45^ and have a “less stressful journey” ^21^. These feelings were the opinion of 14 individuals (7%) who preferred the canal to the road, getting “a wee break from the traffic” ^4^, which can be “so bad in the city” ^95^.

Participants actively used the canal “for getting around” ^130^. More specifically, six (3%) specifically mentioned using the canal instead of public transport. Using the canal “beats sitting on the bus” ^36^ while another experienced feeling “irate by the time you get to work” ^183^ when using public transport while walking along the canal gave them the chance to “get everything sorted in your brain by the time you get to your work” ^183^.

Two participants mentioned the stress they experienced while driving, with one commenting that the canal allowed them to “get an endorphin release” ^95^, which was “good for my mental health” ^95^. Similarly, participants preferred running along the canal to running on the road. Cyclists also enjoyed experiencing the canal as it allows them to “concentrate on my cycling and just free up my mind to my own thoughts…if there were cars around then I’d just have to concentrate on the cars” ^158^.

### 3.2. Activity Space

The canal is a hub for both active and passive recreation.

#### 3.2.1. Active Recreation

Active recreation included using the canal paths for walking, running, and cycling and the canal itself for water sports. Within intercept surveys, participants commented on the importance of physical activity in maintaining and improving good health; “I consider myself fairly active, and I think this is the secret, just keep moving!” ^10^; “I think it’s good to exercise on a regular basis” ^180^. Two particularly noted the importance of doing physical exercise, mainly walking, “especially if you’re old” ^108^ and “aged” ^48^.

The majority of participants actively used the canal either daily (38%) or more than three times per week (24%) (Figure 5).

Modern lifestyles have created a disconnect from the outdoor natural environment, and people now spend significantly more time indoors. However, participants were confident that the canal “definitely keeps you fit, it gets you out and about” ^70^; “It gets me out of the house and to lose a bit of weight. I’ve lost two stone so far” ^128^; “It helps me a lot, it gets me out and about so, other than that, I wouldn’t be out” ^35^. Others commented generally saying, “it does encourage activity” ^182^ and is “obviously beneficial physically” ^1^; “it’s good exercise” ^101^ and helps “get rid of this belly” ^22^ which they said while laughing and patting their stomach, a nod to the importance of physical activity in achieving a healthy weight.

Many canal users actively use the canal as a space for running, walking and cycling, and they made the connection between this and their health. Although we observed people kayaking and canoeing, it was not possible to stop them for an intercept interview, so their perceptions are absent from this research. Sixteen participants (8%) spoke about using the canal to go running, and 21 (10%) used it for cycling. As well as being a space for humans, “the canal is good for the dogs, it helps her (the dog) out” ^188^. Two participants commented that since losing their dogs, they are “very seldom down here” ^72^. Others did not own dogs themselves but enjoyed interacting with them on the canal.

#### 3.2.2. Passive Recreation

In addition to being a space of active recreation, the canal also lends itself to passive activities; it is “quite relaxing just coming to sit by the canal” ^75^. The built infrastructure surrounding the canal can affect how it is used across the life course. We found that older participants sought additional seating to improve their canal experience, allowing them somewhere to rest on their walks. Three individuals discussed fishing along the canal, finding it “peaceful, especially when you’re fishing—it takes off the stress” ^163^, “You get some nice fish here too: pike, perch and roach” ^105^.

### 3.3. Social Space

#### 3.3.1. The Everyday Blue

Participants demonstrated the canal as a place of embodied social interaction, positively affecting their social health. Fundamental to this was the accessibility of the canal. Glasgow is accessible, with pedestrianised streets and transport links including trains, public buses and the Glasgow Subway. Participants commented on the benefits of the canal being “inter-locked” ^8^ with the River Kelvin, allowing them to join up their walks along the canal with other green and blue spaces. The canal is a short walk from Glasgow city centre, and Cowcaddens Subway Station is nearby, making it accessible to other parts of Glasgow. Figure 6 is a map of participants’ postcodes, clearly showing that most canal users live in Glasgow. Although the canal is central and accessible, our data shows that users of the canal predominantly lived hyper-locally, with 45% of participants living within the postcodes directly surrounding the canal.

The canal has become a part of the everyday routines of many participants, taking a pivotal role in their daily schedule: “I jog along the canal in the morning, and that really helps me to start my day, just relaxes me before I go into work.” ^13^. Another appreciated having “a quiet place with some nature near where I live…that’s why I come here every day” ^57^. Others commented on making time in the morning and at the end of the day: “Being able to get out of the city first thing in the morning and after work at night is an excellent way to wind down and relax you and is very therapeutic” ^177^. Thirty-three participants (16%) commented on the canal’s proximity to their home.

As well as being somewhere participants visited before and after work, participants also used the canal during their workday: “I’m here on my lunch break and it just gives me the freedom to think.” ^117^ A local artist worked from a studio near the canal: “I regularly come out and just do a bit of work or email someone on the canal. So it helps me a lot.” ^46^ Another commented, “I work at the Whisky Bond so sometimes I’ll go out for a run at lunchtime, and that helps to alleviate the stress, and I cycle to work and home, so that helps too” ^22^. At the same time, a healthcare provider commented, “if we’ve had a morning of appointments, you can be treating kids all day, and you just need some space, but also you could be at a desk all day, and you just need to move around and get fresh air” ^47^, highlighting that regardless of the type of work being done, the canal can provide a place to unwind and be physically active.

#### 3.3.2. Social Interaction

Participants valued the social interactions experienced on the canal: “It’s a good place to come with…one of my best mates… just showing him around” ^139^, while another enjoyed walking with colleagues on a lunch break to escape the office.

While many individuals used the canal on their own, taking time to escape from the busyness of daily life, many older, lone individuals enjoyed the spontaneous interactions they had with strangers: “meeting people—everybody says good morning. It makes you feel a bit better when everyone is smiling in the morning” ^161^. Another older participant resonated with this, saying, “I suffer from depression and anxiety, and the canal is a good way of getting out the house. It’s a nice way of meeting people, they’re all friendly and chatty and it’s nice—it’s like being in the countryside” ^43^. The comparison made to the countryside is worthy of attention, as it reinforces the capacity of urban blue spaces to create calm environments with a rural feel, both in terms of the natural surroundings, the pace of life, and the friendliness of people.

Four participants (2%) commented on feeling safe while using the canal, especially during the day when the weather is good, and the canal is busy; “seeing more and more people up here using it makes you feel that it is used by the community and you’ll feel safer” ^8^. They highlighted the cyclical nature of canal usage, saying that “the more people using it makes more people use it” ^8^.

#### 3.3.3. Intergenerational Activity

The canal also provides a space for intergenerational activity. Several participants commented on spending time with their family by the canal: “it keeps my children happy, so it keeps me happy” ^157^ while an older participant enjoyed taking her grandson out and staying active ^113^.

#### 3.3.4. The ‘Other’

While some participants commented on feeling safe along the canal when it was busier, the presence of different types of people led to health-limiting safety concerns. The canal can feel “a bit dodgy” ^139^. Two participants commented on the types of people who use the canal who cause it to feel unsafe, using the derogatory term “junkies” ^89, 200^ and “certain characters” ^88^. These phrases are often used to describe homeless people and/or those suffering with drug and alcohol addictions. Other participants talked about the “rumour, culture, idea that lots of stabbings take place” ^68^ and “if you come here alone at night you’re going to get robbed” ^29^. Our study took place in daylight hours, and so many participants noted positive daytime experiences, while many were quick to associate negative connotations with being on the canal at night.

Litter and a lack of bins was mentioned by 23 participants (11%); “The canal is full of rubbish” ^105^ with “loads of dog poo everywhere” ^56^, which detracted from the scenic beauty. A further said that an issue with bins could be that they would not be able to be collected easily by pickup lorries ^85^. One participant made a plea for people to carry their litter with them and then “go home and put it in bin” ^200^. They argued that people have to take personal responsibility for this, saying that “there’s a sense that someone will come and get it”. Three participants commented on taking an active role in minimizing the impact of litter saying “if I have my own gloves and bags with me then I just can’t help myself!” ^86^. Two participants called for people to take more personal responsibility saying, “I think people should respect the canals more; keep it tidy, keep it nice” ^185^.

### 3.4. Symbolic Space

#### 3.4.1. History and Heritage

The historical significance of the canal positions it as a symbolic space. Participants were keen to share their knowledge of the history of the canal and the surrounding area, saying that the more you know about the history, the greater “the feeling” of the area; “once you learn about the history, it does actually change how you think” ^199^. Another participant commented on the “fantastic history…how it was built, who built it, and the history of Mary Hill” ^86^. They commented on the “lovely story” of Miss Mary Hill, who was left the Gairbraid estate by her father, Hew Hill. The land was not profitable until the Forth and Clyde Canal was built, which attracted industry to the area and brought Mary and her husband Robert Graham much-needed money. Mary was immortalised in the area, now named Maryhill. When places are named after people, as is the case of Maryhill, a more profound connection can be made with individuals in history, affording a sense of identity and belonging [48].

Interestingly, the industrial connections were not the only histories that participants mentioned. Participants also described how the canal was managed during World War II, explaining how the canal was a target for the Luftwaffe due to its elevation above the city and the sheer volume of water it held. Bombing the canal would have resulted in flooding, which would have been catastrophic for Glasgow. As a result, temporary barricades were built along the canal to mitigate potential flooding. The decorative signposts installed along the canal were noted by one participant and serve as a reminder of the canal’s history and bridge historic Glasgow with the present day.

As well as collective histories of the canal, participants reflected on their personal histories: “I’ve used the canal since I was a boy, we used to play and swim down here” ^111^. Another was keen to “come back and investigate my family ties to the canal” ^134^.

#### 3.4.2. Restoration

In total, 116 participants (58%) mentioned feelings of restoration afforded by the canal. These were broadly categorised in four ways; using the canal improved mood, was relaxing, allowed participants to switch off and gave them time to think (Table 1).

#### 3.4.3. Health as Holistic

Thirteen participants (6%) explicitly recognised this holistic understanding of their health: “there is a saying you never come back from a walk in a bad mood, and it’s absolutely true” ^148^, “doing exercise is obviously connected to wellbeing so yeah it’s a positive thing for me, absolutely” ^14^. Cycling supported participants’ mental health too: “if you’ve got issues then you can just go on the bike and not think about anything else” ^176^. One participant described themselves going “stir crazy” ^64^ if they do not get out on the canal to exercise, while another poignantly noted, “it’s good for the soul as well as the body” ^87^.

## 4. Discussion

We used Völker and Kistemann’s (2011) extended therapeutic landscapes conceptual framework to understand the Glasgow Canal as a therapeutic urban blue space. The canal was predominantly considered health-promoting, and findings point to the beneficial impact of the canal as an experienced, activity, social, and symbolic space [6].

### 4.1. Experienced Space

Participants benefited from the variety of experiences afforded by the canal, including naturalistic affordances such as fresh air, water, surrounding nature, wildlife, and the changing weather and seasons, and built environmental affordances, including the quietness which results from the absence of traffic.

Urban waterways and the dense vegetation that often frames them are instrumental in supplying fresh air in cities [49]. Air quality affects everyone’s health, although some groups are more vulnerable than others [50]. Emerging research suggests that air pollution can significantly intensify the risk of death from SARS [51] and COVID-19 [52]. Improving air quality may be essential in reducing the spread and impact of air-borne viruses in the future, and blue and green spaces may play a part in this.

Participants noted being energised by the canal. Just experiencing nature has been found to increase feelings of energy [53,54]. Similarly, some participants enjoyed the sounds of water, corroborating an earlier study which found that participants who showed high stress before their experiment experienced bodily relaxation from listening to natural sounds, including running water [55]. Most participants were complimentary of the water, with only one participant commenting on its darkness being disconcerting. A previous study found that users preferred clear water to murky canal water, which has sediment suspended and is often shaded, so it does not reflect the blue sky [7]. Water’s appearance also induces different feelings; water can go from feeling light, bright, and cheerful while reflecting the sun to dark and shadowy, which can be disconcerting [56].

While the main feature of the canal space is the canal itself, a blue space, the term ‘green space’ has been accepted more widely into everyday non-academic language, so it is not surprising that the canal was described as such. Moreover, blue space is often subsumed under the term ‘green space’, but has now been valued as requiring discrete attention [57]. Many languages recognise this blurring of green and blue, colexifying the words; they are expressed using a single term. In Scots Gaelic, ‘gorm’ means ‘blue-green’ and is generally used to describe natural colours in the landscape. Additionally, ‘uaine’ is a bright, vivid green but is also used in names for water environments where the water is coloured by algae or strong mineral content [58]. A reflection here may be that there is a need for ‘blue space’ to describe water environments to be popularised within our everyday language to highlight its significance, independent of green space.

The weather and changing seasons contributed to participants’ experiences of the canal, paralleling previous research which found that participants felt they were happier and more outgoing in the summer months, while in the winter, their mood seemed lower, and they felt lazier and more depressed [59]. The weather has been commonly described as a barrier to outdoor environments and physical activity [9,15]. However, a notable finding in our study was that participants enjoy being outside in the rain as it added to their experience and further connected them with nature. The sound of rain has a regular, predictable pattern that creates calm feelings. Weather changes can also affect peoples’ relationship with nature, and weather can be both comforting while sometimes being difficult and threatening [60]. Our findings support that weather is sometimes absent in literature surrounding nature and wellbeing, but understanding human experiences of weather is fundamental in understanding how people experience nature.

Interactions with non-human actors affected how people experienced the canal. Interacting and feeding wildlife is enjoyable for people as they feel closer to nature and build trust with the wildlife, although there are also negative impacts resulting from these interactions [61]. Additionally, plant and bird species richness can positively affect mental health and wellbeing [15].

Many participants enjoyed the quietness of the canal space. This contrasts with previous research, which described the blue spaces under study as noisy environments which were health-limiting [39]. Noise is ubiquitous in urban living, and the availability of peaceful, quiet areas is decreasing in cities [62]. Noise exposure leads to many auditory and non-auditory health problems, including sleep disturbance and increased risk of hypertension, cardiovascular disease, and cognitive performance in children [62]. Traffic noise has been classified as the second most influential factor contributing to ill-health in Western Europe, behind air pollution [63]. The absence of traffic on the canal lets people declutter their thoughts in a quiet, less stressful natural environment while experiencing the physical health benefits of walking, running, and cycling without worrying about surrounding hazards. Additionally, the ability to take time out of the busyness of city life is of particular importance for people with disabilities, including deaf people, who commonly experience concentration fatigue caused by the heavy concentration required to carry out day-to-day activities, including lip-reading, signing or focusing on listening for an extended period [17]. Access to quiet spaces in urban areas such as the Glasgow Canal may mitigate such negative health consequences.

### 4.2. Activity Space

The canal was used for active and passive recreation, including walking, cycling, running, fishing and sitting. The linearity of canals positions them as subtly different from other urban blue spaces. Similar to findings for linear green spaces, many of the beneficial experiences of the Glasgow Canal emerged from moving alongside its course. In this regard, the canal can be considered a mobile health asset. Most participants reported actively using the towpath by walking, running or cycling alongside the canal. Such findings support the addition of ‘Activity Space’ as a critical dimension within the conceptual understanding of therapeutic landscapes [6]. Walking is a popular form of physical activity, often considered easy, requiring little skill or resources [64]. We found that while some people used the canal to commute, others simply enjoyed taking an aimless stroll, correlating with the understanding of walking as being a highly sensual and complex activity, not solely a transport mechanism or a form of exercise [65]. While some walked alone or with others, some walked their dogs. Domestic dogs and cats are accepted as respected members of today’s society and co-exist with humans in domestic spaces [66]. Evidence has suggested that dog owners engage in more walking and physical activity than those who do not own dogs [67]. The canal provides dog walkers with a space to maximise these health benefits.

Older participants, in particular, appreciated being able to sit, rest and take in the surroundings. Outdoor environments which support passive activities such as sitting by providing benches may benefit the physical health of individuals as they have to travel to reach these spaces of rest [68]. Additional passive activities, such as fishing, took place on the canal. Therapeutic fly-fishing programs for veterans with combat-related disabilities have been found to reduce PTSD symptoms, perceived stress, depressive symptoms and functional impairment and increase leisure satisfaction [69]. Fly-fishing can be meditative with the rhythmic patterns of casting back and forth [70].

### 4.3. Social Space

The canal afforded participants with an everyday social space, where people interacted with one another but were distrusting of others.

Most participants (62%) used the space every day or at least three times a week, highlighting the canal as a routine, everyday blue space instead of a destination for one-off or episodic visits. The frequency of visits allowed people to enjoy the seasonal variation and encouraged people to ‘notice nature’; actively tune into and note the good things in nature. The majority of participants were hyper-local, living less than 1 km from the canal. Therefore, the canals are precious local assets that facilitate and prompt these simple everyday encounters with nature, and by actively noticing things in nature, people may feel more connected to it [71].

Some participants took work breaks by the canal with colleagues as it was close to their workplace. Previous research observed higher reported wellbeing and mood of office workers who walked for 20 min in a blue space than those who walked in an urban environment [72]. Taking nature breaks away from the workplace contrasts with the ‘new normal’ of working from home triggered by the COVID-19 pandemic, where it can be more challenging to separate home and work life. Having local natural spaces to escape to is valuable for those working in traditional workspaces, so it is important that home workers also take breaks in natural settings.

The intergenerational socialisation afforded by the canal space was valuable to some participants. Similar to previous studies, we found the canal allowed older people to interact and socialise, promoting independence and improved mood and wellbeing [73,74]. Our findings support the literature and the concept of Active Ageing, which advocates that promoting opportunities for older people to remain active sustains their quality of life, health, and wellbeing [75]. Our findings about older and younger people spending time together on the canal align with previous studies, which found that observing children can foster restorative benefits [38,76].

Participants predominantly focused on the health-enhancing social interactions afforded by the canal. However, a few participants expressed concern for ‘undesirable’ canal users. The fear of ‘the other’ assumes that homeless people and those with addiction are dangerous to the general public and are considered ‘outsiders’ and excluded from society [77]. We found clear distinctions between day and night perceptions, with night-time being viewed as more dangerous, echoing other studies [38,78,79].

Litter was deemed a health-limiting aspect of the canals, similar to other research on blue spaces [7]. Research of 9757 individuals across the USA found that an absence of receptacle and the distance to the nearest receptacle were both positively predictive of littering [80]. However, the concept of environmental citizenship, and particularly ‘hydro-citizenship’ emerged as some individuals felt a collective responsibility to look after their local landscape [26]. Perceptions of ‘the other’, as well as vandalism and litter, may reflect broken window theory, which states that signs of anti-social behaviour create urban places that encourage further disorder [81]. This has been found to be true of litter; people tend to litter where they see other litter has accumulated [80]. By tackling ‘lesser’ issues on the canal, such as littering by cleaning up existing litter, providing bins, and running awareness campaigns, it is possible that a sense of order may be created which dissuades people from further anti-social behaviours [80]. Blue spaces must be welcoming places for all, and all users’ safety should be considered when designing and redeveloping them.

### 4.4. Symbolic Space

Heritage, histories and the restorative benefits of the canal shaped it as a symbolic space. The histories of places influence people and their communities, but the background to why this occurs is complex [82]. Historic places symbolise community and belonging, ‘the feeling’ of a place and the canal’s history teaches us about the past, uniquely shaping and changing our perspective on present-day Glasgow. Urban blue spaces act as affective place triggers to specific memories and identities. Where memories are created is very important for later recall, a phenomenon known as ‘contextual-binding theory’ [83]. The theory also suggests that individuals are more likely to remember memories in similar contexts [83]. People shared their memories of growing up around the canal while actively using it, corroborating this. Place identity is also apparent as the canal evokes particular social and cultural connotations for individuals, encouraging people to resonate with strong working-class ideologies that go back to its construction.

Everyday stress is a growing concern in cities [84], but there is acknowledgement in the literature that exposure to nature can reduce the risk of stress. The restorative benefits afforded by the canal were discussed by over half of the participants, who reported that the space improved their mood and provided a place to clear their heads and switch off, relax and take time to themselves to think. Nature can provide increased physical and mental distance from stressors and help people become more adaptive and resilient [85]. The Glasgow Canal allowed people to distance themselves from everyday concerns, an affordance also found in previous research on blue spaces [39].

### 4.5. Strengths and Limitations

This study’s main strength is the large sample size, with 203 people being interviewed along the canal towpath across three sites, allowing for an in-depth analysis of experienced health and wellbeing benefits of urban blue space. Although the three sites had nuanced differences regarding landscape typologies, responses did not reflect these differences; instead, there were similar responses across the three sites. To our knowledge, it is the largest study to explore urban blue spaces as therapeutic landscapes. The qualitative, verbatim data provide rich descriptions of how canal users perceived the canal and its ability to impact their health and wellbeing. We acknowledge that this study is not without limitations. We only spoke with people actively using the canal, and so this may have led to an overemphasis on the health-promoting factors; perhaps others in the local area fear the canal and would view it as health-limiting, hence their absence. Stretches of the canal may often be ‘under-seen’ as parts are lined with dense vegetation and so some participants may not know how to access it. Future research should consider exploring in-depth the views of irregular and non-users of the canals to ensure any future developments meet the needs of all local people. Additionally, a survey of a broader geographical sample on population perceptions of the canal could create a more comprehensive picture. Furthermore, peoples’ health and emotions are complex and are affected by many variables, so it is not easy to isolate the impact of environmental exposures [3]. A further limitation is that interviews were conducted during the autumn and winter and it is possible that results may have differed in other seasons. However, given the poorer weather conditions in Glasgow at this time of year, perhaps therapeutic benefits are underestimated as in summer, the space may be better used. Finally, our findings may not be generalisable to all urban blue spaces, given that we only studied one, inner-city post-industrial canal. A similar study conducted in a suburban or rural environment, or in a different country may also have different findings.

## 5. Conclusions

To ensure urban blue spaces are fit for recreational purposes, they must be prioritised as healthy spaces and receive ongoing investment to keep them well-maintained and safe for all users. Ways this can be achieved include improved signage to increase awareness of the connectedness of paths, better lighting to encourage occupation across the day and improved perceptions of safety. Lighting is an example of urban wellbeing infrastructure that can increase feelings of safety and accessibility.

While this research was conducted in 2019, it is worth reflecting on what the findings mean for us today as we continue to be affected by the COVID-19 pandemic and move towards a recovery phase. While the ever-changing lockdown restrictions and the fear of the disease undeniably affected mental, physical and social health, many people have sought refuge in nature. Scottish Canals noted a marked increase in the number of people using the towpaths during the lockdown periods, especially in warmer weather [86]. An online survey of 2115 people in England in the summer of 2020 found that those who partook in the government’s recommended levels of physical activity (150 min) were more likely to report better wellbeing and connectedness to nature, while those who did less than 30 min were less happy and more anxious, highlighting the need to consider green and blue spaces as a critical part of COVID-19 recovery efforts [87].

In increasingly densely populated urban environments, exposure to nature is valued more than ever. Most areas of high deprivation do not have accessible green spaces. However, waterways run through most urban centres. Urban blue spaces, such as canals, can provide therapeutic properties that may help alleviate urban life’s stressors and promote health and wellbeing as experienced, activity, social and symbolic spaces. Our study has uncovered the Glasgow Canal as an everyday therapeutic blue space to be valued as a local health asset.

## Figures and Tables

**Figure 1 ijerph-19-15018-f001:**
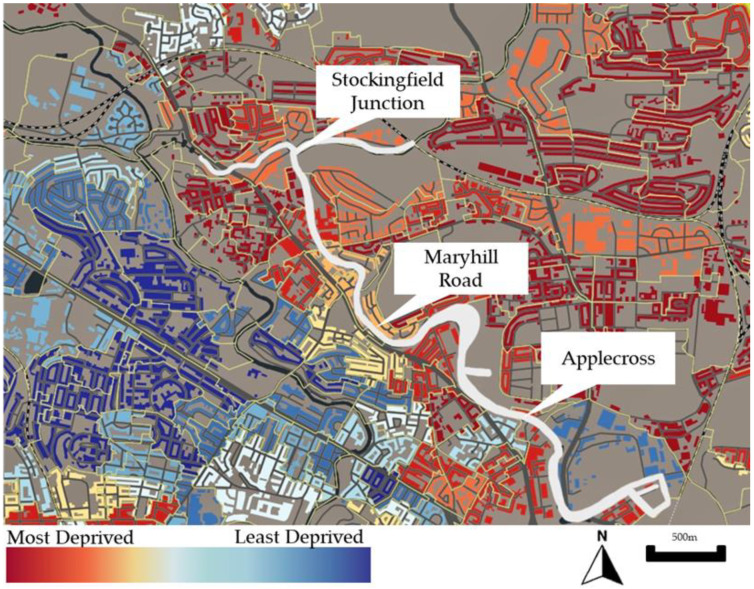
Annotated map of North Glasgow showing Scottish Index of Multiple Deprivation and three study sites (SIMD). Original 2020 map available at www.simd.scot (Contains public sector information licensed under the Open Government Licence v3.0” (from https://www.nationalarchives.gov.uk/doc/open-government-licence/version/3/)) (accessed on 3 September 2021).

**Figure 2 ijerph-19-15018-f002:**
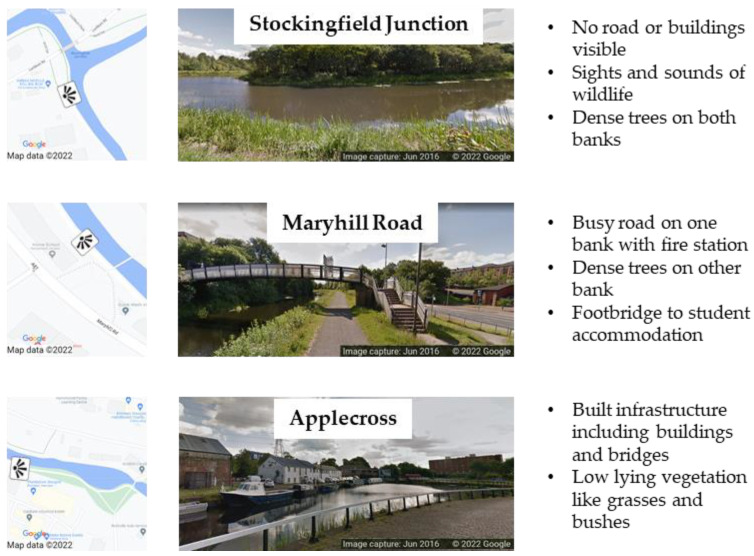
Annotated image of each study site to provide the reader an opportunity to have their own perception of the space. Maps and images © Google, 2022.

**Figure 3 ijerph-19-15018-f003:**
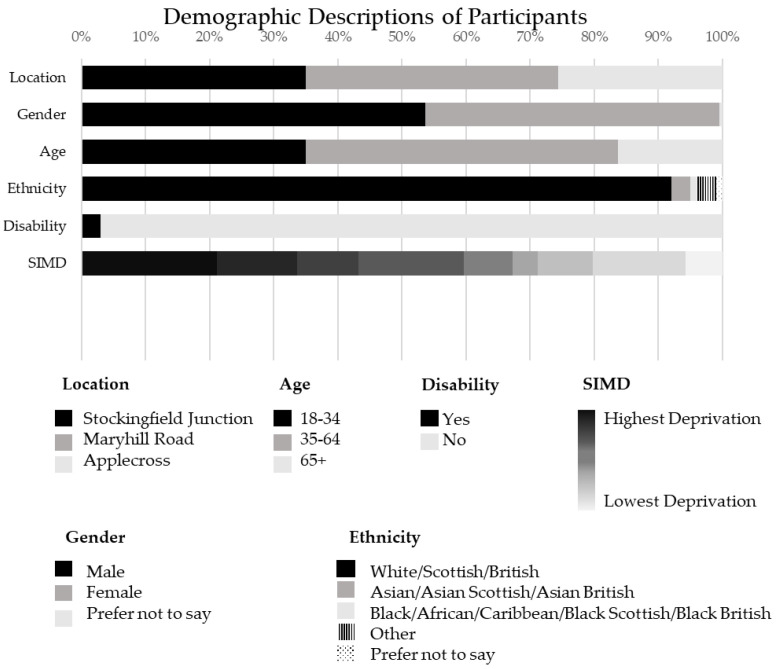
Demographic descriptors of participants [46].

**Figure 4 ijerph-19-15018-f004:**
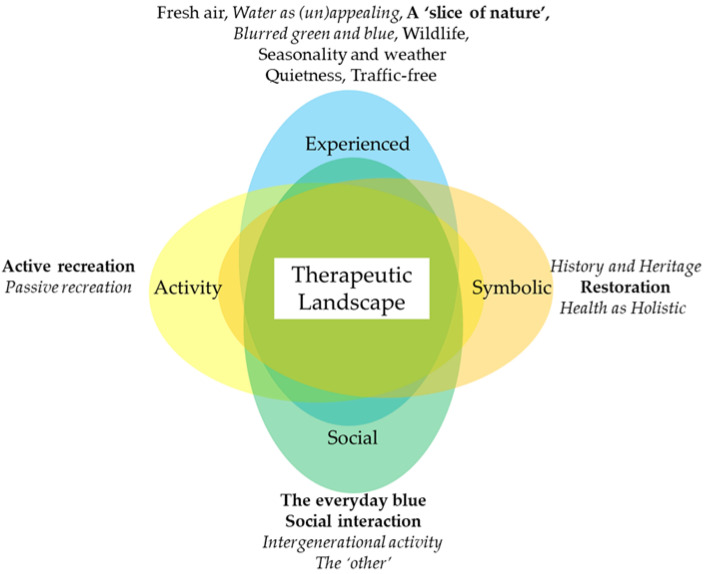
Inductively identified themes categorised within Volker and Kistemann’s (2011) conceptual framework of therapeutic landscapes. *Italicised* themes mentioned by fewer than 10 participants, while **bold** themes mentioned by over 25 [6].

**Figure 5 ijerph-19-15018-f005:**
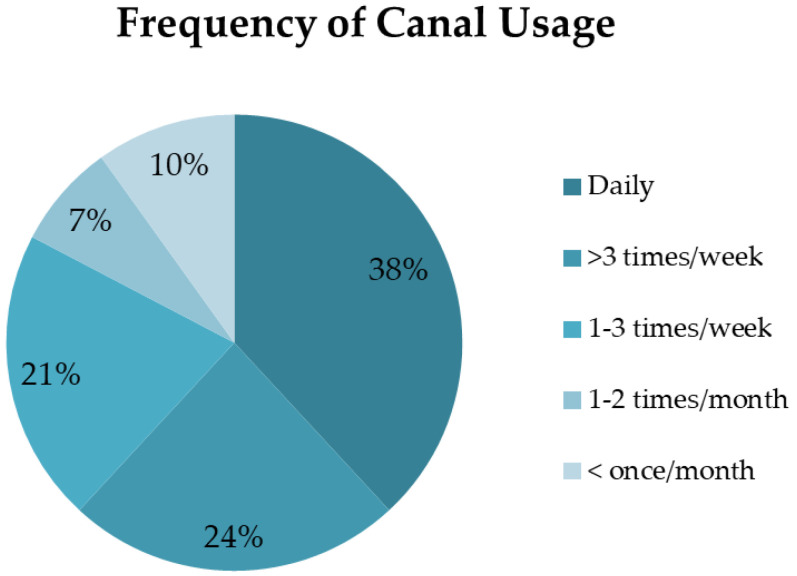
Number of times participants visited the canal (N = 203).

**Figure 6 ijerph-19-15018-f006:**
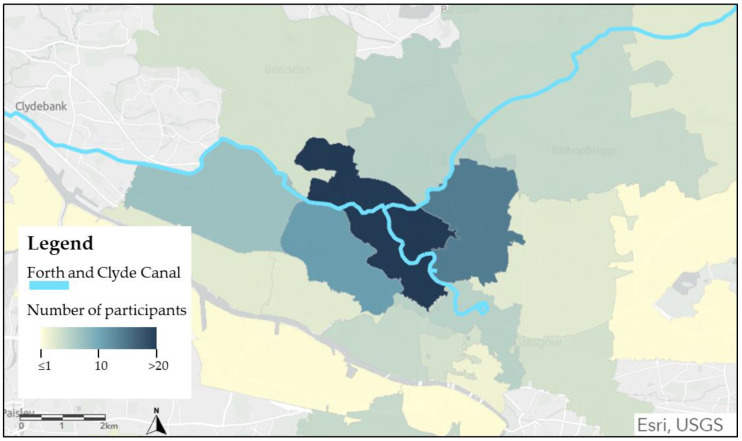
Map of North Glasgow showing the number of participants from each postcode. Map designed in Esri ^®^ ArcGIS online, 2022.

**Table 1 ijerph-19-15018-t001:** Participant quotations on the restorative and mental health benefits of blue space.

Restorative Benefits	No. Participants	Examples
Improved mood	43	“gives me a lift” ^107^ “always makes you feel invigorated” ^87^ “boosts you up” ^83^
Relaxation	41	“very calming” ^121^ “relaxing pace” ^111^ “peaceful” ^30^ “soothing” ^193^
Switch off	18	“get away from troubles” ^34^ “take your mind off things” ^75^ “clear your head” ^43,127,135,159,164,168^ “switch off” ^45, 183^
Time to think	14	“come here and think about things” ^67^ “makes me think about things more, so it’s good for my mental health” ^168^

## Data Availability

The data that support the findings of this study are available on request from the corresponding author, NS. The data are not publicly available due to their containing information that could compromise the privacy of research participants.

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
