# Peer review of "Urban Blue Spaces as Therapeutic Landscapes: “A Slice of Nature in the City”"

_ijerph, 2022, doi:10.3390/ijerph192215018_

Round 1

Reviewer 1 Report

This article proposes the urban blue spaces as therapeutic landscape with the case study of Glasgow Canal which may foster therapeutic properties and contribute to healthier urban environments. The work of this article is clear and logical. However, I have to reject it because of the following problems:

1. The expression of the paper is not clear enough. There is too much narration, lacking of comparative analysis, especially the more expressive graphic analysis and icon analysis.

2. The paper collected data at three access points, but the analysis does not express the differences of plots, or which factors can affect the results. Obviously, the natural conditions of the plot, surrounding land, SIMD, etc. will affect the results, and these influencing factors are also meaningful.

3. The literature review of therapeutic landscape is insufficient.

Reviewer 2 Report

I am grateful for the opportunity to review this manuscript. It covers an important and under-researched topic as urban blue spaces (eg canals) do have great potential for health enhancement. Overall, I think it is well written and nicely utilises relevant theories and frameworks. However, my main critiques are round the structuring of the findings for better clarity and ease of reading/interpretation. Additionally, by following the COREQ checklist and providing greater detail around the methodology (based on comments below), this would improve the manuscript further. With these changes, the paper should be considered for publishing. 

1. Introduction: Great link of this topic to affordance theory as it's highly relevant to this topic

2. Introduction (lines 76-79): another threat is the the further disconnection between people and nature 

3. Introduction: The authors repeat the part about how blue spaces can be risky and health limiting on lines 90-91. Suggest to keep this here and delete where it's mentioned earlier in the Intro (lines 38-39). It fits best in this later part on page 3.

4. Introduction: Good link of research topic to the Volker and Kistemann framework and Gesler and Kearns theory

5. Methods: Great figure that nicely depicts Volker and Kistemann's Therapeutic Landscapes Framework

6. Methods (lines 225-226): Intercept interviews are becoming increasingly common. I would instead reframe this sentence to make clear that they are not necessarily commonplace but are starting to be used more frequently in academia. Here are some example studies (title provided below): 

What entices older adults to parks? Identification of park features that encourage park visitation, physical activity, and social interaction

Critical factors influencing adolescents’ active and social park use: A qualitative study using walk-along interviews

Exploring Children's Views on Important Park Features: A Qualitative Study Using Walk-Along Interviews

Social and Physical Environmental Factors Influencing Adolescents' Physical Activity in Urban Public Open Spaces: A Qualitative Study Using Walk-Along Interviews

7. Methods: Can the authors expand upon the steps used for performing the framework analysis (lines 244-245)

8. Methods: Were participants compensated for their time?

9: Methods: I highly encourage adherence to the COREQ checklist as their are several items on this list that were not included in the Methods section. 

COREQ (Consolidated Criteria for Reporting Qualitative Studies) - Guidelines for Reporting Health Research: A User's Manual - Wiley Online Library

10. Methods: Did all authors agree on coding framework and categories when analysing the interviews? It appears the data were verbatim, not coded? More explicit detail would be good regarding the analysis. 

11. Methods: Can the authors provide more specifics about the sampling? Why were only people actively using the canals approached. Did the authors consider recruiting people in areas near the canal but not necessarily those actively using it? This may have lead to a greater diversity in views as it sounds like their could be bias towards users' of these spaces. It would be good to capture views of people who are non-users of these spaces. Can the authors comment on why this was not part of the methodology? It was noted in the limitations so this was good at least. 

12: Methods: It would be good to specify the season and weather conditions when the interviews were conducted.

13. Methods: Did the authors consider asking more than one open-ended question for the interviews? Why was it just one and why was this most appropriate? Was their prompting involved?

14: Methods: What software was used for analysing the data?

15. Findings: It'd suggest structuring the findings so that key themes are highlighted, followed by quotes. In the quotes, it should be noted the gender, age group of the participants and participants' frequency of canal use. For example something like...

Many participants that they liked the quiet place of nature afforded by the canal:

Quote (participant characteristics)

Quote (participant characteristics)

Or you could have a figure like Figure 6 for each category (symbolic space, activity space, social space, environmental space)

16. Findings: Can the authors provide more specifics about the sample? How many were male vs female, the age groups, demographics, etc?

17. Findings: The authors provide a lot of detail about participants views, but it would be good to make clearer just how commonplace some of the themes were as may sentences indicate views of a single participant.

18. Line 423: What do the authors specify participant 20 when this was not consistently done for other participants when presenting their quotes?

17: Findings: I would also suggest having subheadings to better separate the positive and negative perceptions of of urban blue spaces (ie health enhancing vs health-limiting)

18: I would suggest having a dedicated discussion section. As is, the manuscript is it bit mashed. Under the different categories of spaces, there are positive and negative themes around users' perceptions of these places, mixed with quotes plus links to existing evidence. It would be easier to read and follow if there was a shorter findings/results section and then a Discussion section where findings are linked to existing evidence. 

19: Conclusions: When discussing the limitations, the impact of seasonality and the weather on the dates of the interviews should be noted as it may have influenced the findings. Additionally, only one canal was explored, so the findings may not be applicable to all canals in urban settings. Further, it should be noted that the findings may not be generalisable to non-urban and rural areas, as well as other countries. 
